# First spikes in visual cortex enable perceptual discrimination

**Arbora Resulaj[1,2,3]\*, Sarah Ruediger[1,2,3], Shawn R Olsen[1,2,4], Massimo Scanziani[1,2,3,5]\***

[1]Center for Neural Circuits and Behavior, Neurobiology Section, University of California, San Diego, San Diego, United States; [2]Department of Neuroscience, University of California, San Diego, San Diego, United States; [3]Department of Physiology, University of California, San Francisco, San Francisco, United States; [4]Allen Institute for Brain Science, Seattle, United States; [5]Howard Hughes Medical Institute, University of California, San Francisco, San Francisco, United States

**Abstract** Visually guided perceptual decisions involve the sequential activation of a hierarchy of cortical areas. It has been hypothesized that a brief time window of activity in each area is sufficient to enable the decision but direct measurements of this time window are lacking. To address this question, we develop a visual discrimination task in mice that depends on visual cortex and in which we precisely control the time window of visual cortical activity as the animal performs the task at different levels of difficulty. We show that threshold duration of activity in visual cortex enabling perceptual discrimination is between 40 and 80 milliseconds. During this time window the vast majority of neurons discriminating the stimulus fire one or no spikes and less than 16% fire more than two. This result establishes that the firing of the first visually evoked spikes in visual cortex is sufficient to enable a perceptual decision.

DOI: https://doi.org/10.7554/eLife.34044.001

\*For correspondence:
aresulaj@gmail.com (AR);
massimo@ucsf.edu (MS)

**Competing interests:** The authors declare that no competing interests exist.

## Introduction

Perceptual decisions involve the sequential activation of several, hierarchically organized cortical areas beginning with early sensory areas and ending with associational and motor areas. Based on the number of areas likely involved in the processing of sensory stimuli it has been hypothesized that in each area a relatively brief time window of activity may be sufficient to enable a perceptual decision (*Fabre-Thorpe et al., 1998*). Yet, this time window has never been directly measured for any specific area. By determining these lower limits and analyzing neuronal activity over this time window within a given area we can establish the minimal output of individual neurons in enabling perceptual decisions and reveal how the stimulus is represented within this time frame in that area. Furthermore, this time window defines the time that an area has to be active such that downstream areas can extract sufficient information to enable a perceptual decision. How this time window relates to the time window for an outside observer to extract sufficient information (*Celebrini et al., 1993*; *Mazurek and Shadlen, 2002*; *Shadlen and Newsome, 1998*) is not clear.

The lack of answers to these questions is largely due to technical limitations. One key issue is to demonstrate that the visual area of interest is necessary for the sensory discrimination task at hand. Even though activity in a given area may carry relevant stimulus information, that area may not be required for the perceptual decision. A second challenge is to precisely control the duration of the sensory evoked response of that visual area. Answering this question has been technically difficult since the duration of visually evoked activity in the brain cannot be precisely controlled by the duration of the sensory stimulus. Even a stimulus as brief as 16 ms triggers a response that lasts hundreds of milliseconds in visual cortex (*Rolls et al., 1999*). Presentation of a visual mask at various delays

following the stimulus has been used to perturb the long lasting neuronal response to a visual stimulus (*Kovács et al., 1995*; *Lamme et al., 2002*; *Macknik and Livingstone, 1998*; *Rolls et al., 1994*) and study the effects on perception. However, whether the impact on perception is due to the suppression of the neuronal response to the stimulus or to the generation of the neuronal activity by the mask (*Macknik and Livingstone, 1998*) is difficult to disambiguate. Further, visual masks are not area specific but involve the entire visual system and thus cannot address the minimal duration of activity of a specific visual area. In the mouse, optogenetic approaches make it possible to selectively, rapidly and completely silence neuronal activity of a given brain area (*Lien and Scanziani, 2013*; *Olsen et al., 2012*) at any arbitrary delay after stimulus presentation (*Reinhold et al., 2015*). With optogenetic silencing we do not add activity but instead prevent activity from exiting the silenced area. With this approach we can precisely control the duration of visually evoked activity in a cortical area during a discrimination task.

Here we developed a simple visual discrimination task in mice that depends on visual cortex. By completely and rapidly silencing primary visual cortex at well-defined intervals after the stimulus appeared in the task, we demonstrate that this cortical area is required only during the initial 80 ms from the onset of stimulus evoked response for a reliable decision to be made. Importantly, during this period, most neurons in primary visual cortex fire one or no action potentials. Thus, we establish the minimal time window of activity in primary visual cortex sufficient to enable a perceptual discrimination and provide direct evidence for a key role of the first action potentials fired by individual neurons in the execution of the task.

## Results

### A visually guided behavior that depends on visual cortex

To determine the minimal duration of activity in visual cortex necessary for accurate visual discrimination by the animal, we needed to develop a perceptual task that requires visual cortex. We developed a visual discrimination task in which mice are head-fixed yet free to run on a treadmill (*Figure 1A* and *Video 1*) Visual stimuli (circular patches of gratings,~30 degrees, oriented at different angles) shown on a monitor placed on the right side of the animal moved horizontally from the anterior to the posterior end of the monitor at a speed that was proportional to the running speed of the animal. One of the stimuli (a grating oriented at 90 degrees) was the target stimulus, while the other stimulus (a grating oriented at 45 degrees) was the distractor. Mice were rewarded with water for bringing the target stimulus to the center of the monitor, the reward zone, and holding it there for a minimum time set by the experimenter (~1 s; a trial in which the stimulus is held in the reward zone for at least the minimum time is called a 'stop trial'; see Materials and methods). To start the next trial mice had to bring the stimulus out of the posterior end of the monitor and continue running for some distance. To be most efficient in this task, mice had to continue running when the distractors appeared (*Figure 1B*). To ensure that mice did not solve the task by using local differences in contrast between the two gratings, we varied the position of the stripes in the circular patch, i.e. the spatial phase of the grating, randomly. At the beginning of each trial the stimulus appeared at the anterior end of the monitor and was frozen (i.e. insensitive to the rotation of the treadmill) for 350 ms, after which time the stimulus could be moved by the locomotion of the mouse. This ensured reproducibility of stimulus position across trials in the initial 350 ms. Further, during the task the position of the eye varied little trial to trial during the initial 350 ms (the standard deviation of the position of the right eye over this interval across trials was 2.4 ± 0.6 degrees; mean ±std across five mice; *Figure 1—figure supplement 1*). Mice learned to perform the task with accuracy above 85% correct in 23 ± 7 days (mean ±std; n = 15 wild type mice; *Figure 1—figure supplement 2*; accuracy is defined as the average of the percentage of stop trials upon target presentation and the percentage of non-stop trials upon distractor presentation; chance level is 50%) completing on average 200 ± 30 trials each day (transgenic mice, VGat-ChR2-EYFP, learned the task in 50 ± 20 days, n = 8 mice; difference in learning rates was significant: p=0.0096, Wilcoxon ranksum test, *Figure 1—figure supplement 2*).

To determine whether visual cortex (VC) is required for this visual discrimination task, we used two approaches: optogenetic silencing to determine the impact of an acute and reversible perturbation and surgical lesions to establish the effect of an irreversible ablation. We silenced cortical

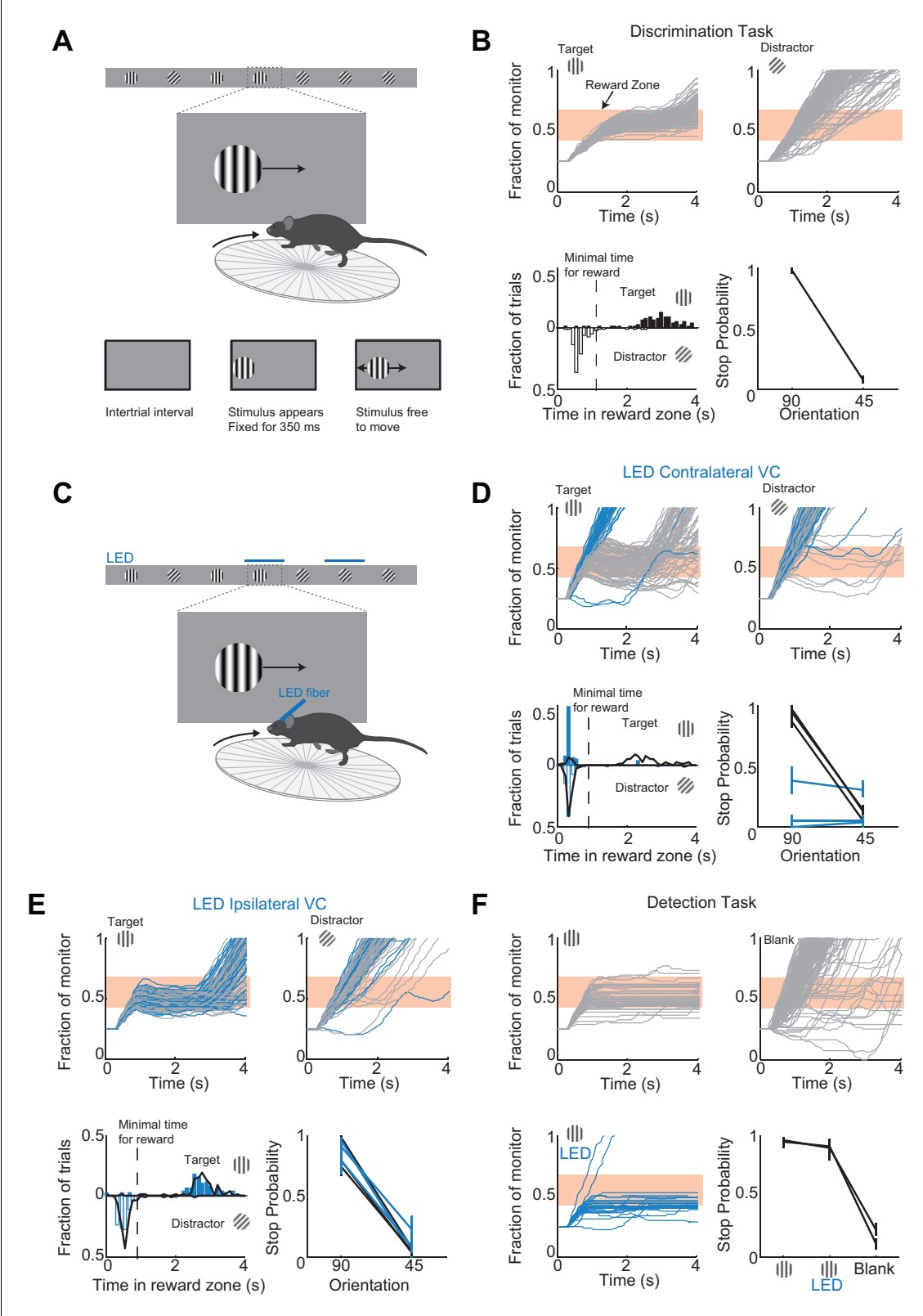

**Figure 1.** A virtual foraging behavior that depends on visual cortex. (**A**) Behavioral setup. The mouse is rewarded for stabilizing the target at the center of the monitor for about a second. (**B**) Example session for a trained mouse. *Top.* Grey lines are individual stimulus trajectories. Orange shaded area is the reward zone. Note different trajectories of target versus distractor stimuli. *Bottom left.* Distribution of the times spent in the reward zone for target (filled bars) and distractor stimuli (empty bars). *Bottom right.* The probability that mice place the object in the reward zone for at least the minimal time

*Figure 1 continued on next page*

*Figure 1 continued*

for reward (stop probability) depends on the identity of the grating. Here and further, error bars are 95% confidence intervals. (C) Behavioral setup as above but visual cortex (VC) is silenced before the appearance of the stimulus and for the duration of the trial on a randomly interleaved fraction of trials. (D) Behavioral performance depends on contralateral VC. Same conventions as in (B). *Top.* Example mouse. Stimulus trajectories during cortical silencing are in blue. This particular mouse systematically overshot the reward zone when centering the target and subsequently moved backwards to bring the target back in the reward zone. *Bottom left.* Distribution of times spent in reward zone. Black: control; Blue: VC silencing. *Bottom right.* Stop probability under control conditions (black) and during VC silencing (blue). Individual lines are individual mice (n = 3). (E) Behavioral performance is not affected by silencing ipsilateral VC. Same conventions as in (D). (F) V1 is not required to express the decision in a detection task. Mice trained with one stimulus only (the target) are rewarded for stabilizing it at the center of the monitor. *Top, Bottom left.* Example mouse. Stimulus trajectories during cortical silencing are in blue. Same conventions as (B). A blank is defined as the absence of a target at regularly spaced distances. *Bottom right.* Stop probability for two mice (individual lines).

DOI: https://doi.org/10.7554/eLife.34044.002

The following figure supplements are available for figure 1:

**Figure supplement 1.** Eye movements in trained mice.
DOI: https://doi.org/10.7554/eLife.34044.003
**Figure supplement 2.** Learning curves for wild type and transgenic mice.
DOI: https://doi.org/10.7554/eLife.34044.004
**Figure supplement 3.** Reversible, rapid and complete silencing of V1.
DOI: https://doi.org/10.7554/eLife.34044.005

activity by optogenetically activating cortical inhibitory neurons (*Atallah et al., 2012*; *Lien and Scanziani, 2013*; *Olsen et al., 2012*) with a 1 mm optic fiber placed over the left primary visual cortex (V1) (i.e. contralateral to the visual stimulus) in transgenic mice (VGAT-mhChR2-YFP) that selectively express the microbial light activated cation channel Channelrhodopsin2 (ChR2) in inhibitory neurons (*Zhao et al., 2011*) (*Figure 1C*). In these mice, V1 activity could be completely, rapidly and reversibly silenced (see *Figure 1—figure supplement 3*) with a delay of 8 ms after the onset of illumination by a blue LED (450–490 nm, *Lien and Scanziani, 2013*; *Olsen et al., 2012*). Cortical silencing started 76 ± 6 ms before the stimulus appeared (mean ±std across mice) and ended just after the stimulus had exited the monitor, and was performed on a third of the trials interleaved randomly. During silencing trials, the behavioral performance of mice was severely disrupted (51 ± 3% accuracy; n = 3 mice; *Figure 1D*). On these trials, mice either kept on running no matter whether the target or distractor was presented (e.g. *Figure 1D*) or, on a fraction of trials, they sufficiently slowed down to center the grating (i.e. stop trial) but did so indiscriminately for both stimuli (p>0.16, Wilcoxon rank sum test on choice data; e.g. *Figure 1D*). Because the distinction between stop and non-stop trials is binary, i.e. based on a threshold duration that the stimulus spends in the reward zone, it is conceivable that while performing at or close to chance when silencing cortex, mice may still hold the target for a longer time than the distractor in the reward zone. For example, targets and distractors may both spend less than the threshold time in the reward zone and hence be categorized as non-stop trials yet the targets may spend a longer time than the distractor in the reward zone. This would imply the ability of the mouse to discriminate despite performing at chance according to the criteria of the task. An advantage of our task is that it can reveal differences in the animal's behavior for target versus distractor that are not captured by the binary classification of stop versus non-stop trials. We thus verified that an ideal observer could not disambiguate the target from the distractor based on times spent by each of the two stimuli in the reward zone using receiver operating

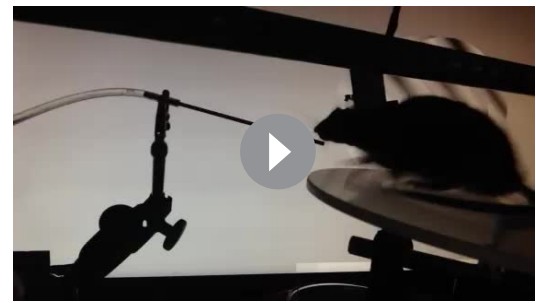

**Video 1.** Video of a trained mouse performing the task. The mouse is rewarded with water for stabilizing the target (90 degree grating) in the center of the monitor. There is no reward or punishment for the distractor (45 degree grating). The speed of the stimulus is proportional to the running speed. Note that the stimulus is frozen (i.e. insensitive to the rotation of the treadmill) for 350 ms following its appearance.
DOI: https://doi.org/10.7554/eLife.34044.014

characteristic (ROC) analysis (see methods). The discrimination accuracy of the ideal observer was 55 ± 6% (mean ±std, n = 3 mice), hence very close to the actual performance of the task. (*Figure 1D*). To exclude the possibility that the optogenetic silencing simply distracted the mice from performing the task, we silenced the right visual cortex (i.e. ipsilateral to the visual stimulus) (*Figure 1E*). This manipulation resulted in no substantial impairment in the behavioral performance (88 ± 6% accuracy for LED trials versus 89 ± 6% for no LED trials; *Figure 1E*), thus showing that the impairment was specific to the visual cortex processing visual information in the contralateral hemifield.

We verified that silencing visual cortex did not affect the ability of mice to express the decision, that is, to place the stimulus at the center of the monitor. Mice were trained as above but with the target stimulus only. In other words, mice where trained to perform a simple detection rather than a discrimination task. The distance that mice had to run to start the next trial was randomly varied. On trials where the contralateral V1 was silenced, mice centered the target image almost as frequently as in control trials (*Figure 1F*) demonstrating that contralateral V1 is not required to express the decision.

Behavioral deficits resulting from acute perturbations of the activity of a given brain area may lead to incorrect interpretations relative to the actual role of that area for behavior (*Otchy et al., 2015*), since following permanent lesions of said area animal's behavior can recover without additional training (*Kawai et al., 2015*). To further assess the necessity of VC in visual discrimination we trained mice to perform the visual discrimination task and, after they had reached proficiency (accuracy of 93 ± 7%, n = 4 mice), we surgically removed VC contralateral to the side of stimulus presentation (e.g. *Figure 2A*) and allowed the animals to recover for ten days post-surgery before behavioral testing. Lesioned animals performed at chance (p>0.3, Wilcoxon rank sum test on choice data, n = 4 mice, *Figure 2B*). The impairment in behavioral performance was not due to the ten day interval from the last behavioral session because trained control animals experiencing even longer intervals between behavioral sessions remained proficient (accuracy of 80 ± 10%, *Figure 2C*). Furthermore the behavioral impairment was not due to either anesthesia or to some unspecific impact of surgery because proficiency was preserved after removing the ipsilateral VC (accuracy of 85%, n = 1 mouse, *Figure 2B*) or following anesthesia to perform craniotomy for physiological recordings (80 ± 10%, see results below). Taken together, these results show that this visual discrimination task requires visual cortex.

## Neurons in primary visual cortex report stimulus identity by 80 ms

To determine over what time interval stimulus evoked spiking activity in individual V1 neurons can be used to disambiguate the target from the distractor stimulus we recorded extracellular action potentials while the animals performed the task (*Figure 3A*). We inserted a multichannel probe in V1 at the beginning of a behavioral session in trained mice (performance accuracy during recordings: 80 ± 10%, mean ± std; n = 9 mice). To ensure that the units were maximally excited by the stimulus, we placed the monitor so that the position of the stimulus in the initial 350 ms, when the stimulus is stationary, was superimposed on the multiunit spatial receptive field (center of stimulus was 2 ± 1 degrees from center of receptive field, mean ± std, n = 8 mice). Further, we compared the eye position during receptive field mapping (performed outside of the task) with the eye position during the task. While the animals moved their eyes more outside of the task than during the task (*Figure 1— figure supplement 1*), the median eye position during receptive field mapping and during the task was very similar (difference of 3 ± 1 degrees; mean ±std across five mice; *Figure 1—figure supplement 1*). For comparison, the size of the receptive field of an individual cortical neuron is 12–20 degrees (*Niell and Stryker, 2008*). Thus, the position of the stimulus was well aligned with respect to the center of the receptive field. The cortical response to the visual stimulus began 40 ± 5 ms after stimulus onset (mean ±std across mice, *Figure 3C*) consistent with previous reports (*Niell and Stryker, 2008*). The onset of cortical response was quantified as the earliest deflection in the local field potential that exceeded three standard deviations from baseline. We verified that the earliest deflection corresponded to layer 4 of V1, the major thalamo-recipient layer, based on current source density analysis (*Niell and Stryker, 2008*) (*Figure 3—figure supplement 1B*).

To determine whether the spiking of an individual neuron allows an ideal observer to discriminate the target from the distractor stimulus we performed ROC analysis (*Tolhurst et al., 1983*) on 72 well isolated units in nine behaving animals (*Figure 3B,D–E*; see Materials and methods for cell type and layer distribution). About half of the neurons (46%) discriminated the target from the distractor when

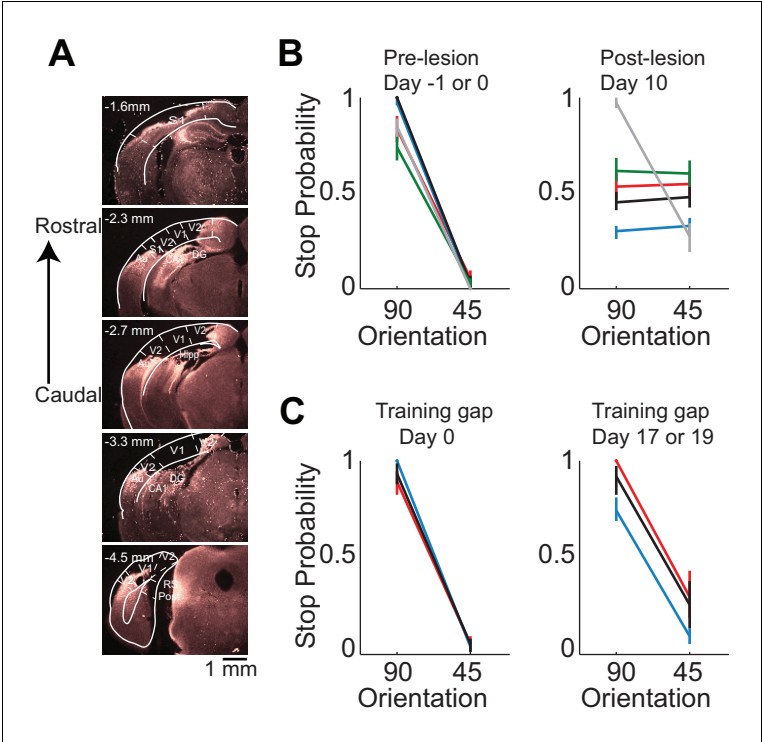

**Figure 2.** Lesion of visual cortex disrupts behavior. (**A**) Coronal brain sections showing the extent of lesion for an example mouse (black in (**B**) and (**C**), 100 μm sections, images of background fluorescence, see Methods for summary of all mice). Outline of the different brain areas is from the Paxinos and Franklin brain atlas (*Paxinos and Franklin, 2007*). Distances are relative to bregma. The retinotopic location in V1 corresponding to the stimulus in the initial 350 ms is ~3 mm from bregma and 2.3 mm from midline. Note the absence of V1 (black). (**B**) Stop probability for the target and distractor stimulus when mice performed the task (*left*) just before the lesion of visual cortex (day 0) or the previous day (day −1) and (*right*) when mice performed the task 10 days after the lesion of visual cortex. Individual lines are individual mice. Each color represents the same mouse in all panels. All mice except the one displayed in grey had lesions in the left visual cortex (contralateral to the stimulus); the one in grey had a lesion in the right visual cortex. Error bars are 95% confidence intervals. (**C**) Stop probability for 3 of the mice in (**B**), (*left*) just before the gap in training (day 0), and (*right*) after the gap in training (day 17 or day 19 without training).

DOI: https://doi.org/10.7554/eLife.34044.006

their activity was integrated over a time window of 300 ms, starting at the onset the cortical response and ending just before the stimulus could be moved by the animal (*Figure 3E*) (p<0.012, Wilcoxon ranksum test on spike counts across trials, Benjamini-Hochberg correction for multiple comparisons). Below we refer to these units as 'discriminating units'. How early do discriminating units start discriminating? We performed ROC analysis at various intervals from the onset of the cortical response (*Figure 3E*). The fraction of discriminating units increased rapidly between 40 and 80 ms (*Figure 3F*). While at 40 ms after the onset of the cortical response only ~20% of the discriminating units discriminated the target from the distractor above chance, by 80 ms already ~50% of units did so with a median discrimination accuracy of 66% (range: 58–79%). The fraction of discriminating units discriminating increased more slowly following these initial 80 ms. By 300 ms (when, per definition, 100% of discriminating units are discriminating) they reached a median discrimination accuracy of 74% (range: 58–96%). Thus, already by 80 ms following the onset of the cortical response ~50% of discriminating units discriminate the target from the distractor.

To determine how well the orientation tuning curve of a neuron predicts its ability to discriminate we measured the tuning properties of discriminating neurons after the end of the behavioral session. We presented drifting gratings of twelve different orientations that had the same size and spatial frequency and were presented at the same location as the stimuli used during the task, yet they

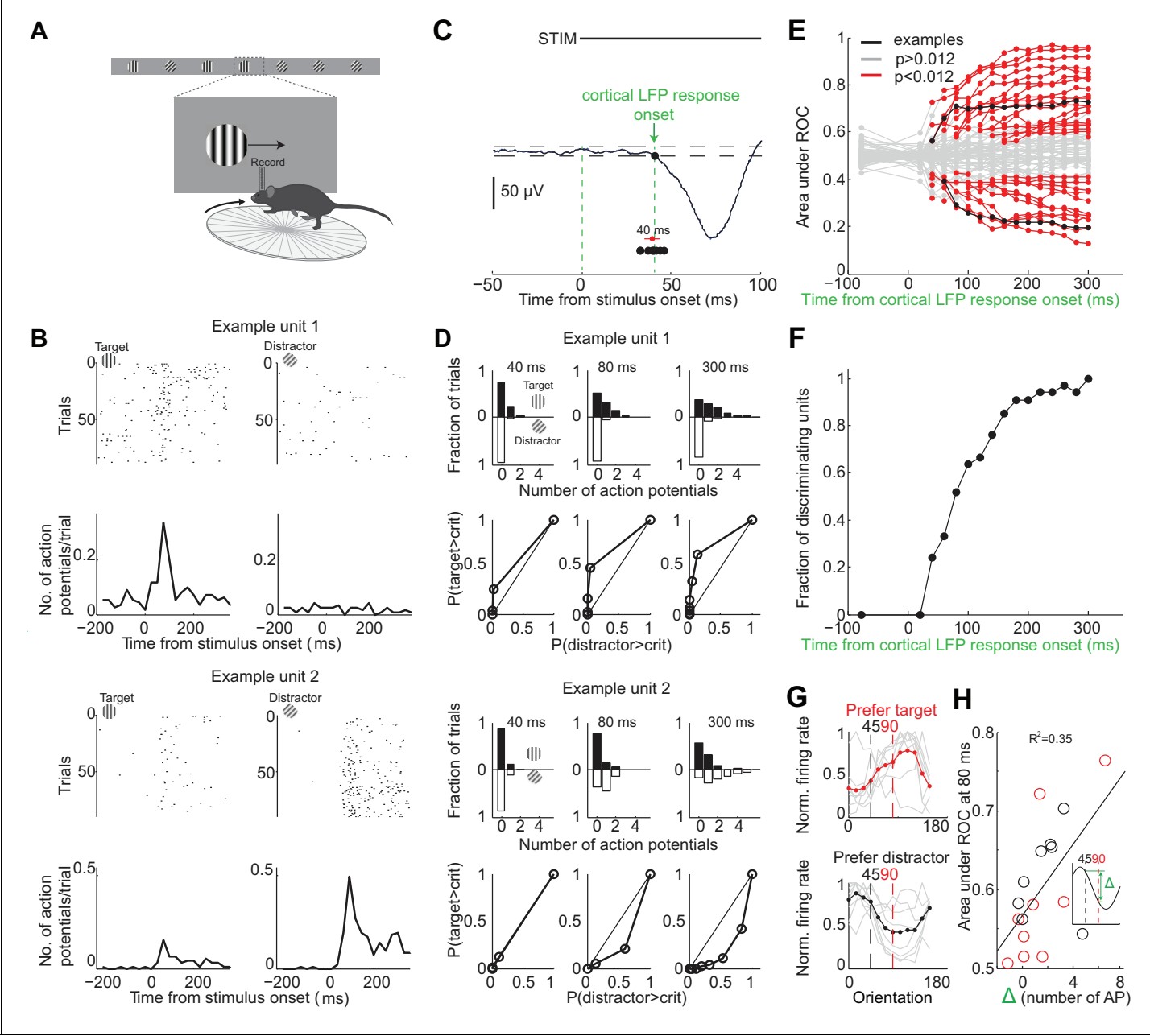

**Figure 3.** Individual neurons can discriminate within 80 ms from onset of cortical response. (A) Experimental setup as in *Figure 1A* but with recording from primary visual cortex. (B) Responses of two example units recorded simultaneously. *Top*. Raster plot. Black dots are action potentials. *Bottom*. Peristimulus time histogram (PSTH). The number of action potentials per trial is calculated in 25 ms bins. (C) Estimation of the onset of cortical response. The onset of cortical response is defined for each mouse as the earliest deflection in the local field potential following stimulus onset. Dashed lines indicate three standard deviations from baseline. Black circles indicate onset of cortical response in nine mice. Red circle and line are the mean and standard deviation across mice. (D) Receiver operating characteristic (ROC) analysis for the two example units in (B). *Top*. Distribution of action potentials across trials for target (black bars) and distractor stimuli (white bars) at three different intervals after the onset of cortical response. *Bottom*. ROC curve for each graph on top. (E) Summary of areas under ROC for 72 units. Area under ROC for individual units (individual lines) depends on the interval from cortical onset. Black: example units in (C) and (D). For each unit at each interval starting at cortical response onset, statistical significance for the separation of the distributions of the number of action potentials for the target versus distractor was assessed using Wilcoxon ranksum test and the Benjamini-Hochberg correction for multiple comparisons (p<0.012). (F) The fraction of discriminating units discriminating at a particular interval increases with increasing time from cortical response onset. A unit is defined as discriminating if by 300 ms p<0.012. (G) Experimental set up as in (A) but stimulus position is always fixed, mouse is not rewarded, and the grating is drifting. Grey lines are orientation tuning curves for individual discriminating units preferring the target (top) or the distractor (bottom) during the task. Colored line is the mean across units. (H) The area under ROC over the initial 80 ms after cortical onset during the task is plotted against the difference in the number of action potentials in response to

*Figure 3 continued on next page*

*Figure 3 continued*

passively viewed stimuli of 45° and 90°. Circles are individual discriminating units (p<0.012, Wilcoxon ranksum test) that prefer target (red) or distractor (black). $R^2$ is the fraction of the variance explained by the linear fit to the data.

DOI: https://doi.org/10.7554/eLife.34044.007

The following figure supplements are available for figure 3:

**Figure supplement 1.** Characterization of V1 recordings during behavior.
DOI: https://doi.org/10.7554/eLife.34044.008
**Figure supplement 2.** Tuning curves for units that do not discriminate.
DOI: https://doi.org/10.7554/eLife.34044.009

were not rewarded and their location was insensitive to the movement of the wheel (passive viewing; stimulus properties: 20°/s; 0.5 s; 15° steps; chosen in a random order; drifting in either of the two directions perpendicular to the grating's orientation). Most discriminating units that preferred the target during the task showed a peak response to orientations larger than 90 degrees (109 ± 8 degrees, median ±SEM; n = 9, four mice; *Figure 3G*). Furthermore most discriminating units that preferred the distractor during the task, showed a peak response to orientations less than 45 degrees (30 ± 20 degrees, median ±SEM; *Figure 3G*; n = 8, four mice). Non-discriminating neurons had either very sharp tuning curves peaking far away from target and distractor orientations, or flat tuning curves, or tuning curves peaking in between the target and distractor orientation (*Figure 3— figure supplement 2*). We compared the difference in spike number in response to grating presented at 45 and 90 degrees during passive viewing with how well discriminating units distinguish the target from the distractor during the task. The difference in spike number during passive viewing correlated with the value obtained from ROC analysis over the initial 80 ms following the onset of cortical response during the task ($R^2$ based on linear fit: 0.35; *Figure 3H*).

## The threshold duration of V1 activity for perceptual discrimination limits most neurons' firing to one or no spikes

What is the minimal duration of activity in visual cortex necessary for accurate visual discrimination? And how many action potentials are fired by individual neurons during this time? If by 80 ms from the onset of visually evoked cortical activity information about stimulus identity is available to an independent observer, it may also be available to the mouse. Thus, the minimal duration of visual cortical activity enabling discrimination may be around 80 ms.

To control the duration of the visually evoked cortical response we optogenetically silenced visual cortex, as described above, at varying intervals after the onset of the response (*Figure 4A*). In each experiment we ensured that the LED intensity was sufficiently high such that performance accuracy was at chance when the illumination started before the stimulus appeared (p>0.05, Wilcoxon ranksum test on choice data, n = 8 mice). Furthermore, as above, for each animal we verified that despite chance performance the hold times of the stimulus in the reward zone of the monitor did not differ between target and distractor stimulus (p>0.05, Wilcoxon ranksum test on stimulus centering times in the reward zone). We verified this again at the very end after testing all LED onset intervals.

The accuracy of the behavior increased with increasing interval between the onset of the cortical response to the stimulus and the onset of cortical silencing (*Figure 4B,C*). When cortical silencing followed the onset of cortical response by 44 ± 6 ms the performance was close to chance (54 ± 5%; mean ±std across mice, *Figure 4D*), similar to when the LED onset preceded stimulus presentation (51 ± 3%; mean ±std across mice). Strikingly, however, when cortical silencing was delayed by a further 40 ms, hence with a latency of 80 ms after the onset of the cortical response, performance accuracy of the animals sharply increased to 76 ± 7% (mean ±std across mice). Performance accuracy continued to increase, yet less sharply, over the longer intervals tested reaching 92 ± 5% when the LED onset followed the onset of the cortical response by 300 ms (mean ±std across mice). With this interval the animals performed similarly to control conditions, in the absence of LED illumination (94 ± 2%; mean ±std across mice). Thus, there is a sharp increase in performance when visual cortex is allowed to function between 44 and 80 ms after the onset of the cortical response. As above, we used ROC analysis to compare behavioral performance with the ability of an ideal observer to disambiguate the target from the distractor based on times spent by each stimulus in the reward zone

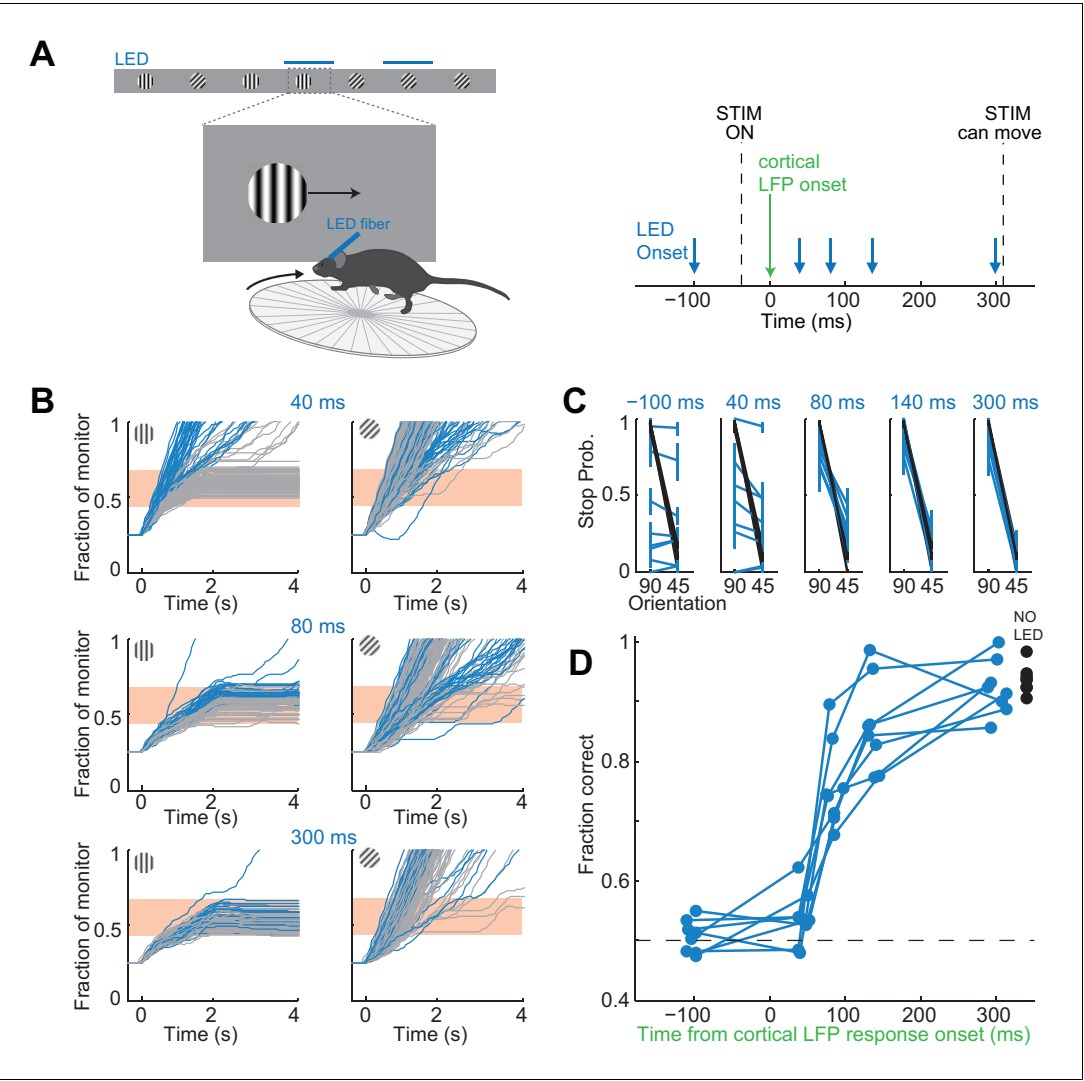

**Figure 4.** It takes visual cortex 80 ms to enable perceptual discriminations. (**A**) *Left*: Experimental setup. *Right*: Arrows indicate the onset of LED illumination. Each interval was tested in separate behavioral sessions. (**B**) Example mouse. Stimulus trajectories during cortical silencing (blue) and under control conditions (gray) for three different LED illumination onset latencies (40 ms; 80 ms; 300 ms) relative to the onset of the cortical response. Individual lines are individual trials. (**C**) Summary of stopping probability for eight mice. Black: control; Blue: cortical silencing. Times indicate the LED illumination onset after onset of the cortical response. Individual lines are individual mice. Error bars are 95% confidence intervals. Note that behavioral performance during cortical silencing increases with increasing LED onset. (**D**) Probability of a correct choice during cortical silencing (blue) depends on the onset of LED illumination relative to the onset of the cortical response. Individual lines are individual mice. Black circles indicate probability correct for individual mice for trials with no cortical silencing.
DOI: https://doi.org/10.7554/eLife.34044.010

when silencing cortex at 44 ms following the onset of the cortical response. The discrimination accuracy of the ideal observer was 54 ± 7%, hence very close to the actual performance of the task at 44 ms (54 ± 5%). These results show that the minimal duration of visually evoked activity in V1 for an animal to perform the present task above chance lies between 40 and 80 ms.

If the estimated time window indeed approximates the threshold duration of V1 activity for perceptual discrimination, performance accuracy in trials when V1 is active for only 80 ms should be very sensitive to the difficulty of the task. We thus trained mice to discriminate a narrower angle difference between target and distractor, namely 15 degrees. Mice were first trained to perform the standard 45 degrees discrimination task and their behavioral performance measured across various

intervals of cortical silencing, as above. We then re-trained those same animals to discriminate a target from a distractor separated by 15 degrees until they reached a similar level of proficiency as for the 45 degrees task (accuracy of 90 ± 4% for 15 degrees versus accuracy of 93 ± 2% for 45 degrees, mean ±std, n = 3 mice, *Figure 5B*). We silenced the cortex of these animals at various intervals following the onset of the cortical response and compared the decrease in performance between the 45 and the 15 degrees discrimination tasks. Silencing cortex at 80 ms after the onset of the cortical response reduced performance significantly more for the 15 degrees as compared to the 45 degrees discrimination task in all animals (p<0.05, Wilcoxon ranksum test on choice data, n = 3 mice *Figure 5B,C*). While silencing V1 80 ms following the onset of the cortical response still enabled the 15 degrees discrimination to occur above chance (p<0.02, Wilcoxon ranksum test on choice data, n = 3 mice), the accuracy was significantly lower than for 45 degrees discrimination (p<0.02, Wilcoxon ranksum test on choice data, n = 3 mice, *Figure 5B,C*). This difference cannot be accounted for simply by a difference in motivation or in control performance because in two out of three mice, non-LED trials during the 15 degrees discrimination task were as accurate as non-LED trials during the 45 degrees discrimination task (Wilcoxon ranksum test on choice data, p=0.65 and 0.67, *Figure 5B*). Thus, these experiments demonstrate that the time between 40–80 ms following the onset of the cortical response indeed captures the threshold duration of V1 activity for a simple perceptual discrimination.

Over these initial 80 ms from the onset of the cortical response discriminating units in primary visual cortex fired only 25 ± 4% of all the spikes fired above baseline during the 300 ms window (mean ±sem across units, *Figure 3—figure supplement 1D*). During this 80 ms interval discriminating units fired 0.6 ± 0.1 (median ±SEM across units) action potentials in response to their preferred

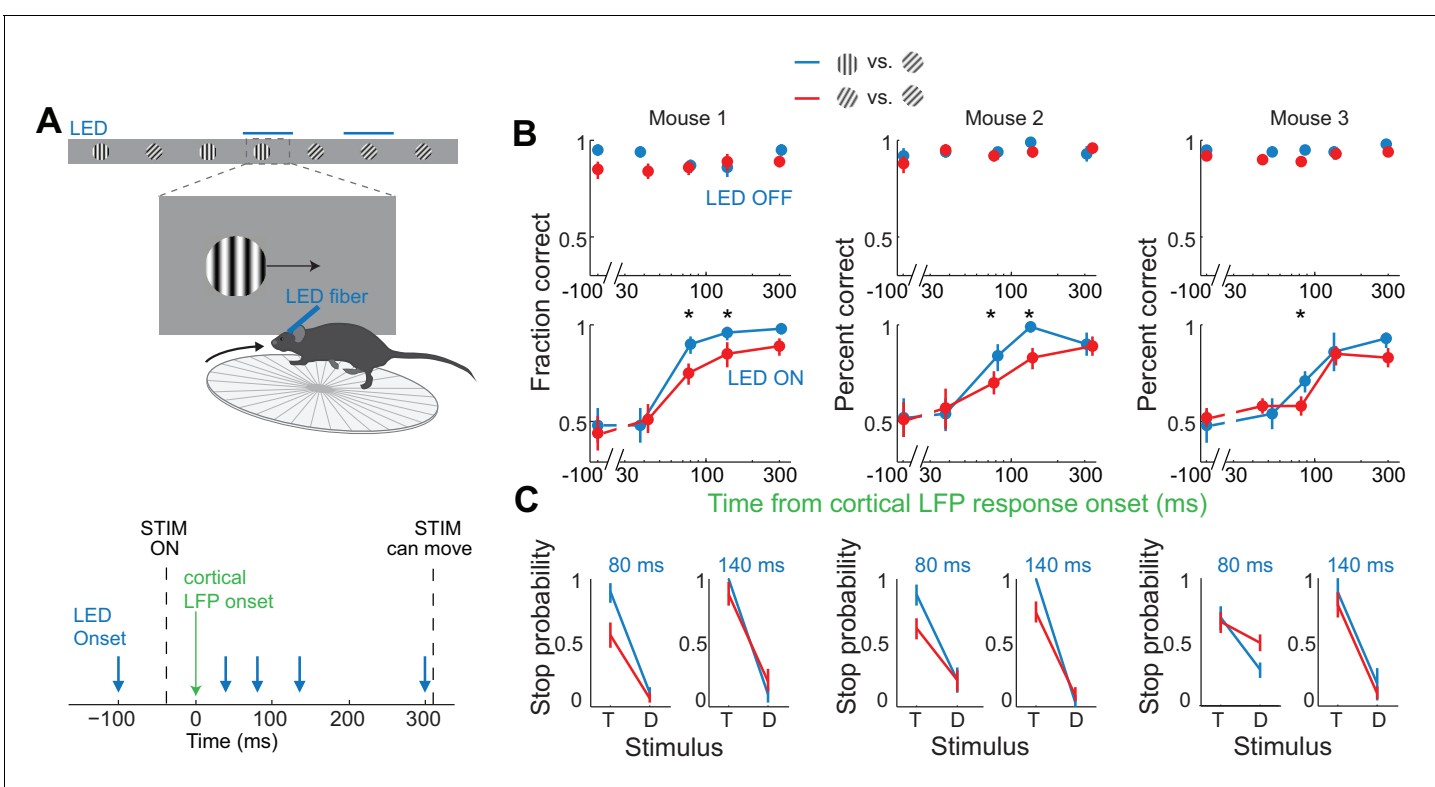

**Figure 5.** Discrimination accuracy when V1 is active for only 80 ms is sensitive to the difficulty of the task. (A) Experimental setup as in *Figure 4A*. (B) Probability of a correct choice for (*top*) control trials and for (*bottom*) trials with cortical silencing for three mice first trained in the main task (blue, target: 90°, distractor: 45°) and then in the harder discrimination task (red, target: 60°, distractor: 45°). Error bars are 95% confidence intervals. Asterisks indicate significant difference in the choice accuracies in the two tasks (p=0.001–0.018, Wilcoxon ranksum test on choice data). (C) Stop probability for the target stimulus (T) and the distractor stimulus (D) for two intervals from (B), *left,* when cortex is silenced at 80 ms or, *right*, 140 ms following onset of cortical response.

DOI: https://doi.org/10.7554/eLife.34044.011

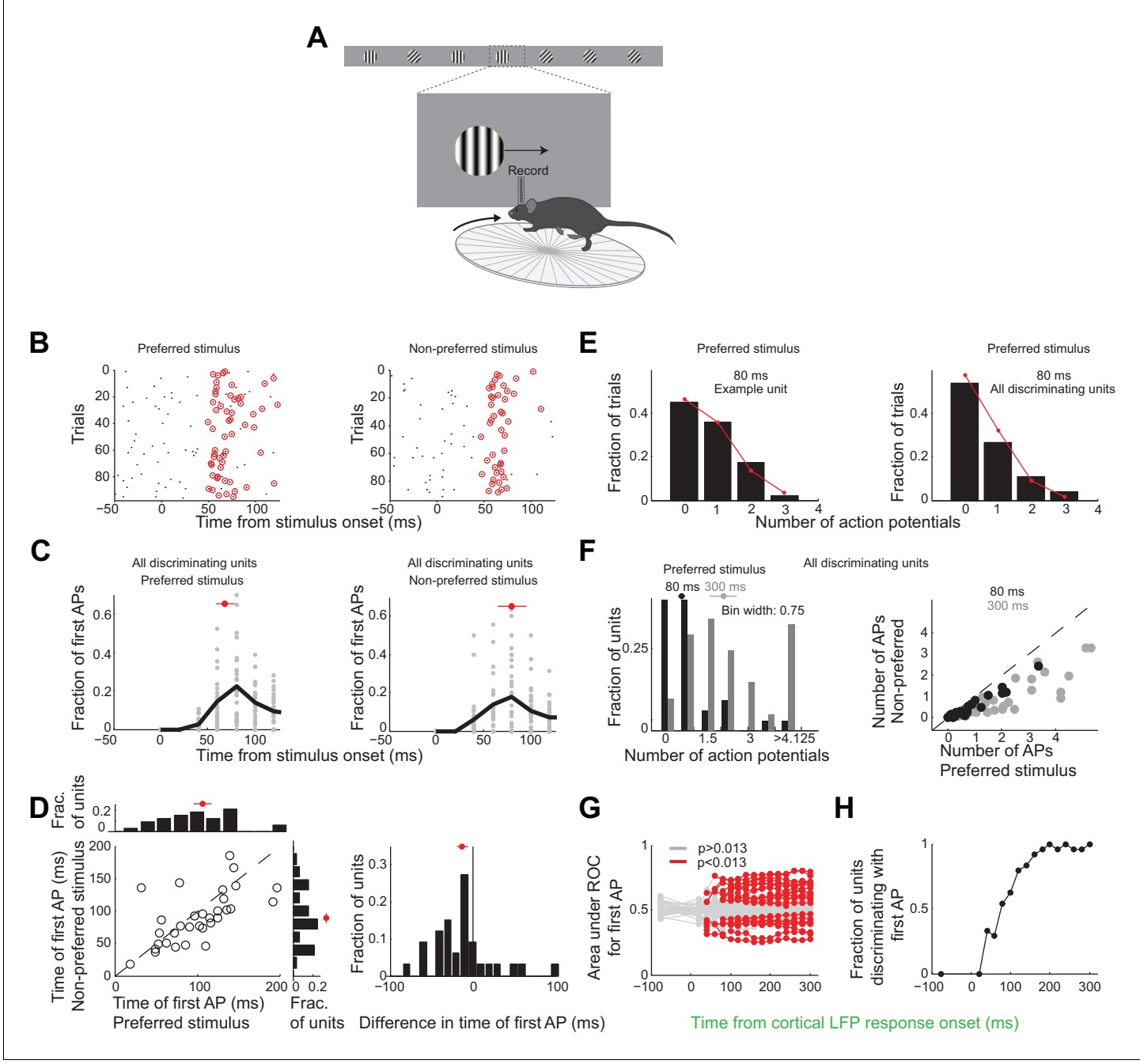

**Figure 6.** Neurons usually fire only their first action potential in the initial 80 ms. (A) Experimental setup as in *Figure 3A*. (B) Raster plot (APs, black dots) for an example unit for preferred (*left*) versus the non-preferred stimulus (*right*). Red circles indicate the first AP in each trial after the onset of the cortical response. (C) The distribution of times of the first AP for individual discriminating units (grey circles, time bins of 20 ms) for those trials in which the first AP occurred within the initial 300 ms following the onset of the cortical response, for (*left*) the preferred and (*right*) the non-preferred stimulus. Black line is the mean across units. Red circle and line through are the median and SEM, respectively. (D) Mean of times of the first AP across trials for individual discriminating units (circles; same trials and window as in C) plotted for the preferred stimulus versus the non-preferred stimulus. *Top.* Distribution of the mean times for the preferred stimulus. *Right.* Distribution of the mean times for the non-preferred stimulus. Red circle and line through are the median and SEM, respectively. *Right panel.* Distribution of the difference in the mean time of the first AP for the preferred and the non-preferred stimulus for all discriminating units. Red circle and line through are the median and SEM, respectively. (E) The distribution of the number of APs across trials for the preferred stimulus for the initial 80 ms after cortical onset for (*left*) the example unit in (B) and for (*right*) all discriminating units approximates a Poisson distribution predicted from the mean number of APs (red line). (F) *Left panel.* The distribution of the number of action potentials (mean across trials) across discriminating units is shown for the preferred stimulus over the initial 80 ms (black) and 300 ms (grey). Last bin also includes units that fired more than 5 APs. Circle and line through are the median and SEM, respectively. *Right panel.* The number of APs (mean

*Figure 6 continued on next page*

*Figure 6 continued*
across trials) for individual discriminating units (circles) for the preferred versus the non-preferred stimulus over the initial 80 ms (black) and the initial 300 ms (grey). (G) Area under ROC for individual units (individual lines), based only on the first AP after cortical onset on each trial plotted against the interval from the onset of the cortical response. Statistical significance was assessed same as in *Figure 3E*. (H) Fraction of discriminating units discriminating depends on the interval from cortical onset. A unit is defined as discriminating if by 300 ms p<0.013.
DOI: https://doi.org/10.7554/eLife.34044.012
The following figure supplement is available for figure 6:

**Figure supplement 1.** Comparison of neural and behavior discrimination.
DOI: https://doi.org/10.7554/eLife.34044.013

stimulus (*Figure 6F*, response not different for the wild type and the transgenic mice as shown in *Figure 3—figure supplement 1C*), corresponding to a firing rate of 7.5 Hz, and 0.22 ± 0.06 action potentials for their non-preferred stimulus. Furthermore, in response to their preferred stimulus, discriminating units fired only one or no action potential in 80% of the trials and two action potentials in only 11% of the trials (*Figure 6E*), similar to what is expected by Poisson statistics (% of variance explained across units: $R^2 = 97 ± 5\%$, median ±SEM; median time until first action potential: 70 ± 10 ms and 80 ± 20 ms from the onset of the cortical response for the preferred and non-preferred stimulus, respectively (median ±SEM across units; analysis performed over the initial 300 ms from the onset of the cortical response, *Figure 6C*); mean latency difference: 12 ± 6 ms (mean ±sem across units; p=0.03; t-test; *Figure 6D*)). Thus, over the initial 80 ms from the onset of the cortical response the vast majority of discriminating units in primary visual cortex get to fire either one or no action potentials.

To determine whether indeed the first action potential in response to a stimulus is sufficient to discriminate the target from the distractor we performed ROC analysis (*Figure 6G*) after removing from each unit all but the first action potential after the onset of the cortical response. As above we performed this analysis for various intervals from the onset of the cortical response. The first action potential was sufficient for ~33% of units to discriminate by 300 ms (compared to 46% if all the action potentials were available), and more than half of those units (54%) could discriminate above chance at 80 ms (*Figure 6H*). Thus for most units the first action potential substantially contributes to their ability to discriminate within the initial 80 ms after the onset of the cortical response.

Finally, the accuracy of the behavioral response during the initial 80 ms can be explained by pooling the activity of ~5 discriminating neurons on average (*Figure 6—figure supplement 1B*), or ~20 neurons if non-discriminating neurons are also included in the pool (*Figure 6—figure supplement 1C*).

Taken together, these results show that the threshold duration of visually evoked cortical activity for a simple visual discrimination lies between 40 and 80 ms, a time window during which most individual cortical neurons get to fire one or no spike.

## Discussion

We have developed a visual discrimination task that necessitates visual cortex because both acute cortical silencing and permanent ablation reduces performance of the task to chance. By silencing visual cortex at various intervals following the onset of the cortical response we show that the lower temporal limits of visually evoked activity for a perceptual discrimination lie within 40–80 ms. The impact on behavioral performance when silencing visual cortex during this time window is particularly sensitive to the difficulty of the task. Importantly, during this initial 80 ms window, most of the neurons in primary visual cortex that disambiguate the identity of the stimulus fire either none or one action potential.

The simple detection of a stimulus can be reported by an animal in response to direct cortical stimulation eliciting not more than one action potential in individual neurons (*Histed and Maunsell, 2014*). Stimulus discrimination via cortical stimulation, on the other hand has been reported only in response to repetitive stimulation eliciting series of action potentials (*Romo et al., 1998*). We show that mice can discriminate visual stimuli even when most neurons in visual cortex are prevented from firing more than their first action potential. Thus, the first sensory evoked spikes of mouse visual cortical neurons are sufficient to drive downstream areas for a reliable execution of the task. This

highlights the ability of cortical areas to instruct downstream targets with only a fraction of their neurons firing a single spike. Similar findings about the essential role of the first spikes have been reported in the early olfactory system (*Resulaj and Rinberg, 2015*; *Wilson et al., 2017*). However, our data also clearly show that extending this time window increases (i) the animal's behavioral performance, (ii) the ability of an ideal observer to disambiguate the stimulus based on the spiking of individual neurons, and (iii) the fraction of neurons that can be used to disambiguate. Extending the time window not only gives more neurons the opportunity to fire their first spike (*Figure 6C,D*), but also enables second and third visually evoked spikes to contribute to the discriminability of the stimulus (compare *Figure 6G* with *Figure 3E*). The extent of this time window for sensory processing is consistent with observations in somatosensory cortex performed during cortical silencing in a detection task (*Sachidhanandam et al., 2013*).

The ability for a neuron to disambiguate two stimuli with only one or no spike depends on how distinct the response of that neuron is for those stimuli and on the trial to trial variability of its responses (*Figure 3G,H*). In mice, visual cortical neurons have orientation tuning functions with relatively broad half widths at half max averaging 30–40 degrees. Given the large trial to trial variability of visually evoked responses in cortical neurons, one may expect that as the difference in orientation between the target and distractor stimuli become narrower, and the overlap in the responses of individual neurons to different stimuli increases, more spikes per neurons, or more neurons spiking may be necessary to disambiguate the stimuli. As a consequence visual cortex may need longer than 80 ms. Consistent with this, our results show that animals trained to perform equally well on a 45 and 15 degrees difference discrimination task, are significantly more impaired on the 15 degrees discrimination task when limiting V1 activity to 80 ms.

Prior work has investigated the minimal time window needed by an outside observer to extract stimulus information from the neuronal activity of an area (*Celebrini et al., 1993*; *Mazurek and Shadlen, 2002*; *Shadlen and Newsome, 1998*). How does the minimal time window needed by an outside observer relate to the minimal time window that is sufficient to enable a perceptual discrimination? Our analysis shows that an independent observer accumulating spikes from around 100 neurons preferring the same stimulus would reliably discriminate the target from the distractor within the initial 30 ms from the onset of the cortical response (*Figure 6—figure supplement 1B*). Because mice discriminate at chance when visual cortex is only active for 40 ms, the minimal time window that downstream areas necessitate to extract sufficient stimulus information is longer than the minimal time window necessitated by an independent observer. This suggests that downstream areas integrate spikes over longer periods than strictly necessary which may either be due to having to overcome noise (*Mazurek and Shadlen, 2002*; *Shadlen and Newsome, 1998*) or unrelated changes, or having to first reconfigure population activity before stimulus information can be integrated. Yet other explanations are possible. Investigating these possibilities will require future identification of downstream areas involved in the task and recordings of neural activity from these areas while visual cortex is silenced. It must also be stated that the time window for the independent observer might be slightly underestimated because neurons were pooled from different experiments and thus the weak correlated noise which exists in simultaneously active neurons, and cannot be averaged out by pooling to increase the signal to noise ratio (*Zohary et al., 1994*), is slightly reduced.

What is the role of visual cortex in perceptual discrimination? Visual cortex is not necessary for all visually guided behaviors in rodents. Several experiment have demonstrated that animals can still perform visually guided behavior even following the silencing or ablation of visual cortex suggesting the involvement of subcortical areas (*Glickfeld et al., 2013*; *Liang et al., 2015*; *Petruno et al., 2013*; *Prusky and Douglas, 2004*). These behaviors however, are either innate or, when learned, enable simple stimulus detection rather than discrimination tasks. Some of these subcortical visual areas may indeed enable our mice to place the stimulus in the center of the monitor while the cortex is silenced, yet other strategies are also possible. At a finer scale, some of our mice showed a bias in the fine positioning of the stimulus within the center of the monitor while the cortex was silenced compared to the non-silenced trials (*Figure 1F* and *Figure 4B*). The reason for this effect is unclear and may depend on the strategy used by our mice to place the stimulus in the center of the monitor while contralateral V1 is silenced. Although we show that visual cortex is required to enable discrimination, consistent with recent work (*Jurjut et al., 2017*; *Poort et al., 2015*), one may debate whether visual cortex plays an instructive role by providing information disambiguating the target

from the distractor to downstream areas or simply a permissive role by regulating the overall excitability of those downstream areas (*Otchy et al., 2015*). We show that permanent ablation of V1 in trained animals reduces task performance to chance levels even ten days following the lesion. This result differs from what is observed after lesioning motor cortical areas on specific motor tasks (*Kawai et al., 2015*). While acute silencing of these motor cortical areas impairs behavior, following permanent ablation of these same areas behavior is regained within a few days without further training (*Otchy et al., 2015*). As a consequence these motor areas are considered permissive rather than instructive for the execution of the behavior (*Otchy et al., 2015*). Instead, given the absence of recovery, our ablation results are consistent with an instructive role of V1. Clearly, we cannot exclude the possibility that the hypothetical area downstream of V1 simply does not recover its original excitability without V1. However, the fact that on the one hand the animal behaves at chance upon silencing cortex before the stimulus presentation, rather than being only partially impaired, and on the other hand that just 80 ms of activity are sufficient to almost completely recover the behavior is further evidence, in our opinion, for an instructive role of visual cortex. The sufficiency of a sensory cortical area to elicit a perceptual decision has been demonstrated for the somatosensory (*O'Connor et al., 2013*; *Romo et al., 1998*) and taste (*Peng et al., 2015*) systems and controlled perturbations of visual areas has been shown to affect decision (*Salzman et al., 1990*) in a predictable manner. The ability to artificially recapitulate the pattern of cortical activity elicited by a visual stimulus through direct cortical activation will eventually provide a definite answer (*Häusser and Smith, 2007*).

It is not clear whether our findings may generalize to the phenomenon of visual masking in which a second stimulus presented shortly after the first may render the first stimulus less visible or invisible. The neural mechanisms underlying visual masking, that is how added activity from two stimuli generates the perceptual illusion of masking, are not well understood (*Breitmeyer, 2008*; *Macknik and Livingstone, 1998*). In the current study we address the simpler and more basic question of the minimal duration of activity in response to a single stimulus still able of triggering a perceptual decision. Indeed there are clear differences between visual masking and the optogenetic approach used here: First, our approach silences neuronal activity while visual masking adds activity (*Macknik and Livingstone, 1998*). Second, our approach is area specific while visual masking impacts the whole visual system. Finally, we can silence visual cortex after most neurons have fired one or no spikes in response to the visual stimulus while, during visual masking experiments, even the just perceivable stimuli elicit multiple spikes (*Kovács et al., 1995*; *Lamme et al., 2002*; *Rolls et al., 1999*). Future experiments using optogenetic approaches may help us understand the neuronal mechanisms underlying the perceptual illusion of visual masking.

We have provided direct evidence for the minimal amount of time that it takes visual cortex to process visual information in order to enable a perceptual decision and determined the neuronal activity that occurs during that period. The speed at which humans are able to discriminate visual stimuli has led to the suggestion that processing of the visual stimuli can be accomplished with individual neurons in each of the relevant brain areas firing either none or one action potential. This work demonstrates that a period of activity in mouse primary visual cortex during which most neurons fire none or one action potential is indeed sufficient to enable perceptual discrimination. Future work will elucidate which downstream brain areas read out these first essential spikes generated in V1.

## Materials and methods

### Animals

All experimental procedures were approved by the University of California San Diego Animal Care and Use Committee (protocol number S02160M). Mice were on a 12 hr light/dark cycle, lights on at eight pm. Training and experiments were performed during the dark cycle. Mice were single-housed. Data were collected from C57BL6 mice (Charles River Laboratories) or for optogenetic silencing, VGat-ChR2-EYFP mice, which have ChR2 targeted to the *Slc32a1* locus (Jackson Laboratories; stock number: 014548). All mice were male and adults (2–5 months old) at the start of experiments.

## Surgery

*Headbar implantation:* Each animal was implanted with a custom made headbar for head-fixation. Briefly, animals were anesthetized with 2.5% isoflurane. Body temperature was controlled by a thermal blanket connected to a rectal thermometer (FHC; DC Temperature Controller). To expose the skull, the skin and periosteum were removed. The gap between the edge of the skull and skin was sealed with Vetbond (Fisher Scientific). The headbar was affixed to the skull with Krazy glue. Dental cement (Lang Dental; Ortho-Jet BCA) was mixed with black ink and applied to reinforce the affixation of the headbar. Animals were allowed to recover for at least 3 days before the start of water restriction (1 ml/day). *Craniotomy*: On the day before the extracellular recording, animals were anesthetized as above and a craniotomy was made over V1 (size: 400 μm x 1 mm, anterioposterior x mediolateral, center approximately 2.3 mm from midline and 1.3 mm from lambdoid suture). At the end of the craniotomy, to protect the brain the craniotomy was covered with a drop of artificial cerebrospinal fluid (ACSF; 142 mM NaCl, 5 mM KCl, 10 mM D-glucose, 10 mM Hepes, 3.1 mM CaCl2, 1.3 mM MgCl2) and Kwik-Cast (WPI). *Cortical Ablation:* The animals were anesthetized as above. Using stereotaxic coordinates the outline of visual cortex (from Paxinos and Franklin mouse brain atlas (*Paxinos and Franklin, 2007*)) was marked at the surface of the skull. Using a dental drill (700 μm) the area of visual cortex was thinned and removed. Sterile PBS was used to hydrate the exposed brain area. A cut of 1 mm depth was performed around the outline of VC using a microsurgical blade (FST). The cortical tissue was removed using a spoon shaped microsurgical blade (FST 10317–14). The area was washed with PBS to remove blood and consequently covered with Silicon Kwik-Cast (WPI). Upon polymerization a layer of cyanoacrylate glue was applied to cover the lesioned area. An additional layer of dental cement was applied to permanently cover the lesioned site.

## Behavioral setup

A schematic of the setup is shown in *Figure 1A*. Mice ran on a custom made flat transparent disc, or wheel (diameter: 15 cm). The wheel was mounted on the shaft of a rotary encoder (MA3-A10-125-B, US Digital), which provided an analog output voltage proportional to the absolute shaft position. The encoder was mounted via an adaptor to a small Noga arm (MSC; part number: 09560459). Data were acquired with a National Instruments data acquisition board (NI USB-6009). To deliver water (~10 μl/reward) we used gravitational flow under the control of a solenoid valve (NResearch; Model 161K011; valve driver: CoolDrive). The valve was connected to a lickspout (hypodermic tubing; gauge 14) via Tygon tubing (1/16 inch ID).

Visual stimulation. Visual stimuli were presented on an LCD monitor (20.5 × 11.5 inches, 1920 × 1080 pixels, 60 Hz refresh rate, gamma corrected mean background luminance: 47 cd/m$^2$ for optogenetic silencing for 6 mice and 120 cd/m$^2$ for two mice, and 110 cd/m$^2$ for electrophysiology). The anterior edge of the monitor was positioned 25 cm from the right eye and the monitor subtended 50 degrees to 150 degrees of the visual field. The monitor was placed on the right side of the animal such that the antero-posterior body axis had a 15 degrees angle relative to the horizontal axis of the monitor with the axes converging rostrally. During the recording, the monitor was moved slightly so that the stimulus when stationary, i.e. in the first 350 ms (*Figure 1A*), overlapped with the multiunit spatial receptive field (see 'Extracellular Electrophysiology'). To quantify how well the stimulus was centered relative to the center of the receptive field, at the end of the recording session we presented black and white squares of 3.6° in a grid of 9 × 9 locations (one stimulus at a time) covering the whole stimulus area for four mice and squares of 6° in a 5 × 5 grid for four mice. The stimuli were generated using PsychToolbox (*Brainard, 1997*) and custom written software in Matlab (Mathworks; code written by Shawn Olsen and available at https://github.com/aresulaj/ResRueOlsSca18; Behavior visual stimulation code and Passive visual stimulation code; *Resulaj, 2018*. Copy archived at https://github.com/elifesciences-publications/ResRueOlsSca18).

For the behavioral task, stimuli were circular patches of static sinusoidal gratings (spatial frequency: 0.146 cycles/degree, diameter: ~30°, contrast: 50%). On each trial, the spatial phase of the grating was chosen randomly out of 7 evenly spaced phases. We monitored the timing of stimulus onset by placing a photodiode (response time 15 ns; PDB-C156-ND; Digikey) at the bottom anterior part of the monitor, where a white square appeared concurrently with the stimulus after a 5 ms delay (accounted for in our analysis). The horizontal motion of the stimulus was controlled by the running of the animal, and updated at ~20 Hz (monitor refresh rate: 60 Hz). The gain, defined as stimulus

displacement on the monitor (cm)/running distance (cm) varied from 0.3 to 0.6 across animals depending on how fast each animal ran (*Supplementary file 1*). The distance between two consecutive stimuli in the track was 1.25 times the width of the monitor.

For the recording sessions, the spatial phase of the grating was constant and did not vary trial to trial. During passive viewing (*Figure 3G,H* and *Figure 3—figure supplement 2*), the stimuli had the same size and spatial frequency and were presented at the same location as the stimuli used during the task during the initial 350 ms, yet the stimuli were not rewarded and their location was insensitive to the movement of the wheel (passive viewing). We presented drifting gratings of twelve different orientations (3 Hz thus 20 degrees/s; 0.5 s; 15° steps; chosen in a random order; drifting in either of the two directions perpendicular to the grating's orientation). The duration of the inter trial grey screen was 0.75 s. We presented ~30 repetitions/direction. The size and spatial frequency of the stimulus as projected on the retina will vary depending on its position on the monitor. However data on electrophysiological recordings only report activity for a specific position of the stimulus on the monitor, when the stimulus is fixed. This position is the same for stimuli presented during the task and during passive viewing.

## Monitoring of eye movements using infrared video-oculography

Eye movements were monitored as previously described (*Liu et al., 2016*). Briefly, we used a high speed infrared (IR) camera (Imperx IPX-VGA 210; 200 Hz) to capture the reflection of the right eye on an IR mirror (Edmund Optics #64–471). Images were acquired using National Instruments PCI-6036E and custom written software in LabVIEW (National Instruments). Requests for eye tracking code should be addressed to Satoru Miura (Satoru.Miura@ucsf.edu), who wrote the code and kindly shared it with us. The pupil was identified by thresholding the pixel values. The eye position was quantified as the distance between the center of the pupil and the center of the corneal reflection of a reference IR LED placed above the camera along the optical axis of the camera. The position was calibrated by swinging the IR LED and the camera by ±10 degrees along a circumference centered on the image of the eye. Because eye movements impair the calibration and our mice move their eyes less during the task than outside of the task, (*Figure 1—figure supplement 1*) the calibration was performed for each mouse during the behavioral task the day before the measurement were taken. Following calibration, the camera was locked in place.

## Photo-activation of cortical interneurons to silence V1

An optical fiber (1 mm) coupled to a blue LED (470 nm; Doric Lenses) was placed over V1 above the intact skull covered with a thin layer of Krazy glue. The fiber was placed at approximately the retinotopic location corresponding to the stimulus during the initial 350 ms in the task: ~2.3 mm from midline and ~1.3 mm from lambdoid suture. To find these coordinates, we recorded multiunit activity in V1 with the monitor in the same position as during optogenetic silencing (monitor was moved <15 degrees to center the spatial receptive field of the multiunit activity). We used these same approximate coordinates for all of our recordings.

For each animal, the total power was increased until the performance was at chance level when the LED illumination started before the stimulus appeared (3.3–20 mW across animals; p>0.05; Wilcoxon ranksum test on stimulus centering times in the reward zone). To turn the LED on at specific delays after stimulus onset, the photodiode signal detecting the onset of the stimulus was sent to an amplifier (Newark; TWLUX - TW-MF2CAB) and then to an external microprocessor (Mega 1280; Arduino). The microprocessor waited for the amplified photodiode signal to cross a threshold before sending out a digital trigger to the LED driver (Thorlabs). The jitter (s.d.) of the LED onset was 4 ms.

## Behavioral training

Training began after animals had been on water restriction for at least 7 days (~1 ml water/day). During training, mice were kept at 80% or above of their initial body weight. Additional water was provided if the body weight fell below 80% of the initial weight. The initial behavioral parameters were: 100% contrast, gain (gain = stimulus displacement in the monitor (cm)/running distance (cm)): 0.6, hold time (minimal time in reward zone for a reward): 0.2 s. With these parameters, mice would get a water reward every time a new target stimulus would pass the reward zone, that is, as long as they kept running. After mice began to run consistently (one to a few sessions), the gain was decreased

to 0.45 and the hold time was initially increased to 0.4 s to get the first ~10 rewards each session (i.e. during the warm up period) and then increased to 0.9 s. Mice learned to perform the task, that is to hold the target in the reward zone for at least the minimal hold time for a reward, but not the distractor, with accuracy >85% in 23 ± 7 days (mean ±std; n = 15 wild type mice) completing on average 200 ± 30 trials each day (transgenic mice learned the task in 50 ± 20 days, *Figure 1—figure supplement 2*). Over this period, if mice were running too fast and not stopping on a substantial fraction of targets, the gain was decreased (lowest value: 0.3). Conversely, if mice were stopping on a substantial fraction of distractors, the hold time was increased (up to a value of 1.5 s). After mice achieved ~85% accuracy, stimulus contrast was decreased to 50%. Mice easily generalized to stimuli of 50% contrast.

The animal's discrimination accuracy based on the binary classification of stop versus non-stop trials could be lower than that of an ideal observer monitoring the time that the stimulus spends in the reward zone. This could be the case if, for example, on some target trials the mouse slows down more than it would for a distractor trial but not sufficiently so for the target to spend the minimal amount of time in the reward zone and hence be classified as a stop trial. This scenario would lead to an underestimate of the animal's ability to discriminate. For cortical silencing, when the LED illumination started before the stimulus appeared, the animal's discrimination accuracy was similar to that of an ideal observer based on ROC analysis of the stimulus centering times in the reward zone. However, when the LED illumination started after the stimulus appeared, particularly for intervals longer than 80 ms from onset of the cortical response, the animal's discrimination accuracy was usually noticeably lower than that of an ideal observer (>10% difference). This difference would often occur because on some target trials mice would not slow down sufficiently for the trial to be a 'stop trial' but they would slow down more than they would for distractor trials. Thus, an advantage of our task is that it revealed differences in the animal's behavior for target versus distractor that were not captured by the binary classification of stop versus non-stop trials.

To motivate mice to make choices with similar accuracy to that of an ideal observer monitoring the time that the stimulus spends in the reward zone, we adjusted the probability that an image would be a target and the minimum time that the target had to be centered for a reward (hold time). Decreasing target probability increases the gap between rewards. With decreasing probability of the image being a target, any miss is accompanied by a longer average time interval until the next target arrives. Similarly, as the probability of the image being a distractor increases, a given false alarm rate increases the average time until the next target is available. The parameters were adjusted until for LED illuminations starting 301 ms after cortical onset (*Figure 3*) mice made choices that had an accuracy similar to that of an ideal observer (difference did not exceed 7%). Adjusting parameters for the longest interval between the onset of the cortical response and the LED illumination (301 ms) was usually sufficient for the discrimination accuracy by the animal to be similar to that of an ideal observer for shorter intervals too. For data collection, parameters were kept constant across different intervals. For a list of final parameters for all mice included in the main experiments see *Supplementary file 1*. The probability that a stimulus would be a target varied from 25–50% across mice, and the hold time varied from 0.6 s – 1.0 s across mice. Each interval was tested for 1–3 sessions totaling 130 ± 70 trials (range: 42–372 trials per interval), and data were pooled together for analysis.

## Extracellular electrophysiology

On the day of the recording the Kwik-Cast was removed, ACSF was added and, before the recording electrode penetrated the brain, a drop of 1% agarose (Type IIIA; Sigma-Aldrich) was added to reduce movement artifacts. The recording electrode was a NeuroNexus 32 channel linear probe (A1 × 32-Edge-5mm-20–177) that span 620 μm in depth across cortical layers. The probe was inserted approximately perpendicular to pia and lowered to a depth of ~850–900 μm (the curvature of the V1 surface was estimated using the Franklin and Paxinos brain atlas (*Paxinos and Franklin, 2007*)). The probe was connected with a Plexon adaptor to two 16-channel A-M Systems headstages (gain 20x) and then connected to two 16 channel A-M Systems amplifiers (Model 3600; gain: 500x, high pass filter: 0.3 Hz, low pass filter: 5 kHz). The voltage signals were acquired with a National Instruments data acquisition board and extracted with custom written software in Matlab (code written by Shawn Olsen and included in https://github.com/aresulaj/ResRueOlsSca18; Electrophysiology Acquisition; *Resulaj, 2018*. Copy archived at https://github.com/elifesciences-publications/ResRueOlsSca18).

Data collection began at least 30 min after insertion of the probe. We first presented black or white squares of ~10° to map the location of the receptive field across all channels of the probe. To ensure that all receptive fields overlapped with the stimulus position during the behavior in the first 350 ms of a trial (*Figure 1A*), we either moved the monitor slightly if the movement was approximately <15°, or reinserted the probe at a different location mediolaterally. We mapped the receptive field at a higher spatial resolution at the end of the recording for eight mice (for same units and same stimulus and monitor position, see 'Visual Stimulation').

## Histology

Mice were transcardially perfused with 4% paraformaldehyde (PFA) in PBS. The brain was post-fixed in 4% PFA overnight in 4°C and then cut in 100 μm thick coronal sections using a vibratome. To estimate the extent of the lesion, consecutive sections were used. All mice had lesions in V1 and surrounding V2 areas. The lesion extended slightly into the following areas (as outlined in the Paxinos and Franklin mouse brain atlas (*Paxinos and Franklin, 2007*)): hippocampus (2/5 mice), retrosplenial cortex (2/5 mice), primary somatosensory cortex (2/5 mice), primary and secondary auditory cortex (1/5 mice), temporal association cortex (1/5 mice), parietal association cortex (2/5 mice), dorsal subiculum (1/5 mice), postsubiculum (2/5 mice).

## Data analysis: behavior

We visualized the positioning of the stimulus on the monitor by plotting the position of the leading (caudal) edge of the stimulus (*Figures 1* and *3*). A stop trial is defined as the stimulus spending ≥the minimal time for reward in the reward zone (500 pixels wide). Error bars in stop probability (*Figures 1*, *2*, *4* and *5*) indicate 95% confidence intervals assuming a binomial distribution of stops and non-stops at each orientation (data pooled from all sessions that each condition was tested). Accuracy for the target stimulus is defined as the percentage of stop trials upon target presentation; accuracy for the distractor stimulus is defined as the percentage of non-stop trials upon distractor presentation. Overall accuracy is taken as the average of these two choice accuracies; chance level is 50% correct.

## Data analysis: spike sorting

We used UltraMegaSort (*Fee et al., 1996*; *Hill et al., 2011*) to sort spike waveforms into clusters. We then manually sorted the clusters into putative units. Units were accepted as well isolated according to the following criteria. First, the spike waveform across four channels had to be different than that of neighboring units in the cluster. If there were similarities, the orientation tuning curves had to be different otherwise the units were merged. Second, the refractory period violations for each unit did not exceed 0.1% (except for *Figure 1—figure supplement 3* where threshold was 0.2%). To reach this criteria, we occasionally removed outliers (*Hill et al., 2011*) identified using the distribution of the Mahalanbois distance of spike waveforms from the cluster center. Third, for each unit, the fraction of spikes with amplitudes below detection threshold (4 s.d. of high frequency noise) did not exceed 15% (including any removed outliers) as estimated by a Gaussian fit to the distribution of spike amplitudes. Finally, we ensured that the spike waveform was stable for the duration of trials in our analysis.

Out of 98 well isolated units (n = 9 mice, one recording/animal), 12 units (12%) were putative inhibitory units based on spike waveform (*Figure 1—figure supplement 3*). Isolated units spanned all cortical layers.

## Data analysis: electrophysiology

To estimate the onset of the cortical response (*Figure 3C*), we first averaged for each channel the raw voltage traces across trials. We then filtered the averaged trace (fourth order Butterworth filter, low pass frequency cutoff: 300 Hz, applied bidirectionally) to get the local field potential. The filtered trace was almost indistinguishable from the raw trace (blue versus black, *Figure 3C*). The onset of the cortical response was defined as the earliest deflection in the filtered traces exceeding 3 s.d. from baseline. The baseline was computed for each channel as the average over the interval −80 ms to 20 ms from stimulus onset.

The current source density analysis based on the averaged traces was computed as described before (**Niell and Stryker, 2008**; **Reinhold et al., 2015**). For the average CSD across mice in **Figure 3—figure supplements 1B**, three mice were excluded because either the recording electrode did not span the same depth of cortex as other recordings (n = 1 mouse) or there was noise in a few channels (n = 2 mice); both cases would lead to discontinuities in the average CSD if included.

To quantify for each recording how far the center of the receptive field was from the center of the stimulus, at each stimulus location (black or white squares in a 9 × 9 grid in 4 mice and 5 × 5 grid in four mice covering the whole stimulus area; see 'Extracellular Electrophysiology') and for each channel we calculated the baseline subtracted response over a window of 170 ms (stimulus duration: 120 ms), and then normalized the responses to the peak response. After, for each location we averaged the responses across all channels. We then computed for each location the average of the average response to the white squares and the average response to the black squares. Lastly, we estimated the center of the receptive field by calculating the center of mass from the average responses at different locations. The distance from the center of the receptive field to the center of the stimulus was 2 ± 1 degrees (mean ±std, n = 8 mice).

We computed the area under the receiver operating characteristic (ROC) using function *perfcurve* in Matlab. To exclude the possibility that this analysis was not sensitive enough for low firing units, out of 98 well isolated units, 13 (all regular spiking) were excluded because they fired <1 spike every six trials over the initial 300 ms. This threshold was chosen because units firing at rates just above this threshold could discriminate (p<0.012; Wilcoxon ranksum test comparing the distributions of the number of action potentials for target versus distractor). We confirmed that the distribution of running speeds for the two stimuli was not significantly different in the initial 350 ms and thus did not affect our ROC analysis (p>0.02, Wilcoxon ranksum test using the Benjamini-Hochberg correction for multiple comparisons, n = 9 mice).

To determine how many neurons are needed to explain behavior, we first artificially increased the number of units by randomly shuffling the trials of each unit to get six new units. We increased the total number of units to 231 units for the pool containing discriminating units only and 504 units for the pool containing both discriminating and non-discriminating units. We then randomly picked N units, where N was 2, 5, 10, 20, 50, 100, or 200. For all units preferring the distractor, we switched the target responses with the distractor responses. Next we created a 'pooling neuron' that had for each trial all the spikes of all N neurons. We then performed ROC analysis on the spikes of this pooling neuron. We repeated the random sampling step 1000 times and averaged the resulting areas under the ROC curve. We repeated the whole procedure 10 times. Error bars in **Figure 6—figure supplement 1** are sem from these 10 repetitions.

To compute the time of the first spike for the preferred versus the non-preferred stimulus (**Figure 6B–D**) we quantified for each unit the time of the first spike for each trial over the initial 300 ms following the onset of the cortical response. For each unit we then normalized the distribution of these spike times (total number of spikes = 1) and then averaged across units the fraction of spikes in each time bin (20 ms bins, **Figure 6C**). We also show the mean of the spike times for each unit and the distribution of these means across units (**Figure 6D**). To assess whether the first spike occurred earlier for the preferred versus the non-preferred stimulus, for each unit we computed the difference in the mean time of the first spike for the preferred versus the non-preferred stimulus and tested whether the mean of the differences from all units was different than zero (Student's t-test).

To compute orientation tuning curves, the firing rate for each orientation was calculated over the initial 330 ms following cortical onset (i.e. the first cycle of presentation), averaged across repetitions, and normalized by the maximal firing rate across orientations.

To compute the preferred orientation for each unit, we used the following equation (**Lien and Scanziani, 2013**):

$$x = \sum r_k \cos(2\theta_k); \quad y = \sum r_k \sin(2\theta_k)$$

$$\text{Preferred orientation} = 0.5 \times \arctan(y/x) \text{ if x>0}$$

$$\text{Preferred orientation} = 0.5 \times (180 + \arctan(y/x)) \text{ if x>0}$$

We computed orientation selectivity index (OSI) using the equation (**Lien and Scanziani, 2013**):

$$OSI = \frac{\sqrt{(\sum r_k \sin(2\theta_k))^2 + (\sum r_k \cos(2\theta_k))^2}}{\sum r_k}$$

To evaluate the dependence of the area under the ROC curve over the initial 80 ms after cortical onset during the task versus the difference in the number of action potentials in response to passively viewed stimuli of 45° and 90°, we fitted a linear function using least squares estimation (*Dobson and Barnett, 2008*). The standard errors of the slope and offset parameters of the fit were based on the inverse of the information matrix (*Dobson and Barnett, 2008*). The slope was significantly larger than 0 ($p < 0.05$; t-test).

## Statistical analysis

The stated p-values are the results of the Wilcoxon ranksum test unless otherwise noted. For medians, we report standard errors calculated using bootstrapping.

## Acknowledgements

We thank J Evora for help with mouse husbandry, Jeffery Isaacson, Lindsey Glickfeld and Michael Shadlen for comments on the manuscript, Maria Dadarlat and members of the Scanziani lab and Isaacson lab for helpful discussions, Satoru Miura, Guy Bouvier and Baohua Li for help with headbar implantations and eye tracking, and Yi Li for help with behavioral training and histology. We are grateful to the undergraduate students that helped with training of the mice. This project was supported by the Gatsby Charitable Foundation and the Howard Hughes Medical Institute.

## Additional information

### Funding

| Funder | Author |
| --- | --- |
| Howard Hughes Medical Institute | Massimo Scanziani |
| Gatsby Charitable Foundation | Massimo Scanziani |

The funders had no role in study design, data collection and interpretation, or the decision to submit the work for publication.

### Author contributions

Arbora Resulaj, Conceptualization, Formal analysis, Writing—original draft, Writing—review and editing, Performed experiments; Sarah Ruediger, Writing—review and editing, Performed lesions of visual cortex; Shawn R Olsen, Writing—review and editing, Developed the behavioral task; Massimo Scanziani, Conceptualization, Funding acquisition, Writing—original draft, Writing—review and editing

### Author ORCIDs

Arbora Resulaj http://orcid.org/0000-0002-9886-1380
Massimo Scanziani http://orcid.org/0000-0002-5331-9686

### Ethics

Animal experimentation: All experimental procedures were approved by the University of California San Diego Animal Care and Use Committee. (protocol number S02160M).

### Decision letter and Author response

Decision letter https://doi.org/10.7554/eLife.34044.017
Author response https://doi.org/10.7554/eLife.34044.018

## Additional files

**Supplementary files**
• Supplementary file 1. Parameters for the behavioral task for each of the mice included in the main experiments. Hold time is the minimal time that the target stimulus has to spend in the reward zone for a reward to be available. Track gain is the stimulus displacement on the monitor (cm)/running distance (cm). Target probability is the fraction of stimuli that are the target stimulus (stimuli are randomly interleaved).
DOI: https://doi.org/10.7554/eLife.34044.015

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
