## [Decision Letter]

Thank you for submitting your article "First spikes in visual cortex enable perceptual discrimination" for consideration by *eLife*. Your article has been reviewed by two peer reviewers, and the evaluation has been overseen by Inna Slutsky as the Reviewing Editor and Eve Marder as the Senior Editor. The following individuals involved in review of your submission have agreed to reveal their identity: Ilan Lampl (Reviewer #1); Carl CH Petersen (Reviewer #2).

The reviewers have discussed the reviews with one another and the Reviewing Editor has drafted this decision to help you prepare a revised submission.

Summary:

This outstanding paper shows that animals can discriminate between simple visual stimuli based on the early response in the visual cortex, during which neurons fire zero or one spikes at the most. Since cortical response even to brief stimuli lasts far beyond the duration of the stimulus, the time-window which is sufficient for perception of the stimulus has been unclear. Many attempts were made in previous studies to address this question, mostly using computational approaches and psychophysics. However, the question was never directly addressed due to the lack of adequate methods to precisely control the duration of cortical activity. Briefly, Rsulaj at al., used optogenetic inactivation of V1 in mice that performed a novel discrimination task in order to find the minimal time-window of V1 activity that is required for discrimination. By controlling the time of cortical inactivation with respect to the onset of visual stimuli, they show that V1 activity during the first 80 milliseconds from the onset of stimulation is sufficient for discrimination. Extracellular recordings indicated that during this period most cells in V1 fired no more than one spike.

The authors did a superb job in designing the study and the experiments were conducted with the highest standards. The data clearly support the conclusions of this study. The authors thought carefully on their study and included very important controls, which strengthened the conclusions. The presentation of the results is remarkably good and the discussion is clear. Overall, this is a very important study that puts forward clear evidence for the speed of cortical processing that is involved in perception of simple visual stimuli that animals use for making decisions.

Here are our suggestions to strengthen the manuscript:

Essential revisions:

1) The optogenetic inactivation of the contra V1 during detection task is a very important and nice experiment, but it may require additional experiments or changing the statements that appear in the results. Unlike in cats and primates, many LGN cells in mice are driven by both eyes (Howarth et al., 2014, Rompani et al., 2017). Hence, neurons in the ipsilateral V1 are likely to respond to the visual stimulation that was used in this specific detection task and contribute to the animal's decision. Thus, it is well possible that the primary visual cortex is required for detection. To fully address this issue additional recordings from the ipsilateral V1 and bilateral silencing is needed. Yet, since revealing the underlying mechanisms of detection is not a crucial part of this study the alternative way to overcome this difficulty is simply by changing their statements to say that the contralateral V1 is not required for detection in their experimental design. See also the following comment.

2) Moreover, in monkeys LGN cells directly project to higher cortical areas (Sincich et al., 2004). Since such organization might also exist in mice, the authors can only conclude that in the detection task the contralateral V1 is not required. See first comment.

3) It is not clear why in the detection task, inactivation of contra V1 biased the distribution of 'fraction of monitor' towards smaller values compared to the non-inactivated trials. This clearly suggests that inactivation of the contralateral V1 in detection task affects the behavior and should be discussed.

4) The conclusions of this study are somewhat dependent on the assumption that the position of the stimulus relative to the receptive field of the recorded cells is well aligned. However, if the animal makes strong eye movements during the task (when the stimulation appears) other (non-recorded) cells could be activated. Thus, it will be important to check the position of the eyes during this task. While such data will have no effect on the strong conclusion regarding the duration of cortical activity that is needed for discrimination, it is important for the APs data.

5) We think the authors should discuss two closely-related studies from the mouse whisker system in which the timing of neuronal activity in wS1 was studied. Sachidhanandam et al., (2013) found strong suppression of performance when wS1 was inactivated simultaneously with sensory stimulus, and a reduced effect when inactivation was delayed by 100 milliseconds (although both were significant). Guo et al., (2014) found that inactivation of wS1 during whisker-object sampling was more effective than inactivation of wS1 during a delay period (although both were significant). Here, in the current study, the authors report strong behavioural impairment when V1 is inactivated within ~100 milliseconds of the visual stimulus, with 76% performance for optogenetic inactivation starting at 80 milliseconds after visual cortical response (latency 40 milliseconds) recovering to 92% performance when inactivation is started at 300 milliseconds after onset of the cortical response. The authors results are therefore consistent with previous reports, which would seem appropriate to discuss.

---

## [Author Response]

The authors did a superb job in designing the study and the experiments were conducted with the highest standards. The data clearly support the conclusions of this study. The authors thought carefully on their study and included very important controls, which strengthened the conclusions. The presentation of the results is remarkably good and the discussion is clear. Overall, this is a very important study that puts forward clear evidence for the speed of cortical processing that is involved in perception of simple visual stimuli that animals use for making decisions.Here are our suggestions to strengthen the manuscript:Essential revisions:1) The optogenetic inactivation of the contra V1 during detection task is a very important and nice experiment, but it may require additional experiments or changing the statements that appear in the results. Unlike in cats and primates, many LGN cells in mice are driven by both eyes (Howarth et al., 2014, Rompani et al., 2017). Hence, neurons in the ipsilateral V1 are likely to respond to the visual stimulation that was used in this specific detection task and contribute to the animal's decision. Thus, it is well possible that the primary visual cortex is required for detection. To fully address this issue additional recordings from the ipsilateral V1 and bilateral silencing is needed. Yet, since revealing the underlying mechanisms of detection is not a crucial part of this study the alternative way to overcome this difficulty is simply by changing their statements to say that the contralateral V1 is not required for detection in their experimental design. See also the following comment.

The reviewers are correct that many LGN cells in the mouse are driven by both eyes (Howarth et al., 2014, Rompani et al., 2017). However, these LGN cells respond to the binocular part of the visual field that encompasses -30 to +30 degrees along the azimuth relative to the central meridian (Howarth et al., 2014). Because in our task the monitor is placed on the right side of the animal in the monocular part of the visual field, it is unlikely that these LGN cells are activated during the detection task. Yet, as requested by the reviewers, we have changed our statement to say that the contralateral V1 is not required for detection in response to the reviewers' comment #2 (please see below).

2) Moreover, in monkeys LGN cells directly project to higher cortical areas (Sincich et al., 2004). Since such organization might also exist in mice, the authors can only conclude that in the detection task the contralateral V1 is not required. See first comment.

We thank the reviewers for the comment. In the mouse it is known that anterograde tracers injected in the LGN label faintly the V2 area lateral to V1 (Antonini et al., 1999, Oh et al., 2014). Further, Sanderson et al., showed using retrograde tracing that V2 areas receive <5% of their thalamic input from LGN (compared with 95% for V1; Sanderson et al., 1991). These studies support the conclusion that the LGN projection to higher visual cortical areas is unlikely to play a major role in the detection task. Yet, other subcortical pathways to higher cortical areas also exist (Glickfeld and Olsen, 2017). Thus, we agree with the reviewers that, in the absence of optogenetic silencing of higher visual cortical areas during the detection task, we can only conclude that in the detection task the contralateral V1 is not required. We have modified our conclusion in the text accordingly.

3) It is not clear why in the detection task, inactivation of contra V1 biased the distribution of 'fraction of monitor' towards smaller values compared to the non-inactivated trials. This clearly suggests that inactivation of the contralateral V1 in detection task affects the behavior and should be discussed.

The reviewers are correct that, for the example mouse shown in Figure 1, inactivation of contra V1 biased the distribution of 'fraction of monitor' towards smaller values in the reward zone compared to the non‐inactivated trials. For the other mouse used for the detection task (not shown), inactivation of contra V1 biased the distribution of 'fraction of monitor' towards larger values in the reward zone compared to the non‐inactivated trials. Similarly, for the example mouse shown in Figure 4, inactivation of contra V1 starting at 80 milliseconds following the onset of cortical response biased the distribution of 'fraction of monitor' towards larger values compared to the non‐inactivated trials. In general, we saw a bias within the reward zone in either direction in some of our mice. Therefore, we agree with the reviewers that this clearly suggests that the inactivation of the contralateral V1 affects the fine positioning of the stimulus within the reward zone. We now mention this in Discussion section. The reason for this affect is unclear and probably depends on the exact strategy used by the animal to position the stimulus in the reward zone while contralateral V1 is silenced.

4) The conclusions of this study are somewhat dependent on the assumption that the position of the stimulus relative to the receptive field of the recorded cells is well aligned. However, if the animal makes strong eye movements during the task (when the stimulation appears) other (non-recorded) cells could be activated. Thus, it will be important to check the position of the eyes during this task. While such data will have no effect on the strong conclusion regarding the duration of cortical activity that is needed for discrimination, it is important for the APs data.

We agree with the reviewers that for our conclusion regarding the number of APs, it is important that the position of the stimulus relative to the receptive field of the recorded cells is well aligned. We have now monitored the eye position during the task in 5 mice and found that the position of the eye varied little: the standard deviation of the position of the eye across trials during a window of 80 ms was 2.4 ± 0.6 degrees (mean ± std across 5 mice; Figure 1—figure supplement 1). For comparison, the size of the receptive field of an individual cortical neuron is 12-20 degrees (upper – deeper layers in V1, Niell and Stryker, 2008). Niell and Stryker quantified size as full width at half maximal response of a fitted Gaussian, and thus at the edges of the receptive field a neuron still fires on average half of the spikes relative to maximal activation. We also compared the eye position during receptive field mapping (performed outside of the task) with the eye position during the task. While the animals moves its eyes more outside of the task than during the task the median eye position during receptive field mapping and during the task was very similar (difference of 2.7 ± 0.9 degrees; mean ± std across 5 mice; Figure 1—figure supplement 1). Therefore, given the small difference in median eye position during receptive field mapping and during the task, the position of the stimulus remains on average relatively well aligned during the task.

5) We think the authors should discuss two closely-related studies from the mouse whisker system in which the timing of neuronal activity in wS1 was studied. Sachidhanandam et al., (2013) found strong suppression of performance when wS1 was inactivated simultaneously with sensory stimulus, and a reduced effect when inactivation was delayed by 100 milliseconds (although both were significant). Guo et al., (2014) found that inactivation of wS1 during whisker-object sampling was more effective than inactivation of wS1 during a delay period (although both were significant). Here, in the current study, the authors report strong behavioural impairment when V1 is inactivated within ~100 milliseconds of the visual stimulus, with 76% performance for optogenetic inactivation starting at 80 milliseconds after visual cortical response (latency 40 milliseconds) recovering to 92% performance when inactivation is started at 300 milliseconds after onset of the cortical response. The authors results are therefore consistent with previous reports, which would seem appropriate to discuss.

Indeed, our data are consistent with observations in somatosensory cortex performed during cortical silencing in a detection task by Sachidhanandam et al., (2013). This is now stated and cited in the Discussion. The work by Guo et al., on the other hand addresses time windows that are an order of magnitude longer and hence not directly comparable with the present work.